# Bridges and Vertices in Heteroboranes

**DOI:** 10.3390/molecules28010190

**Published:** 2022-12-26

**Authors:** Stuart A. Macgregor, Alan J. Welch

**Affiliations:** Institute of Chemical Sciences, School of Engineering & Physical Sciences, Heriot-Watt University, Edinburgh EH14 4AS, UK

**Keywords:** boron cluster, heteroborane, bridge, vertex, structure, crystal structure, DFT calculation

## Abstract

A number of (hetero)boranes are known in which a main group atom *X* ‘bridges’ a B—B connectivity in the open face, and in such species *X* has previously been described as simply a bridge or, alternatively, as a vertex in a larger cluster. In this study we describe an approach to distinguish between these options based on identifying the best fit of the experimental {B*_x_*} cluster fragment with alternate exemplar {B*_x_*} fragments derived from DFT-optimized [B*_n_*H*_n_*]^2−^ models. In most of the examples studied atom *X* is found to be better regarded as a vertex, having ‘a ‘verticity’ of ca. 60–65%. Consideration of our results leads to the suggestion that the radial electron contribution from *X* to the overall skeletal electron count is more significant than the tangential contribution.

## 1. Introduction

Boron cluster chemistry, which traces its roots to the seminal works of Stock [1], Lipscomb [2], Hawthorne [3] and many others, is now a mature but still developing field which sits at the intersection of organic, inorganic and organometallic chemistry. From the earliest times it was appreciated that the polyhedral nature of these species meant that the bonding within them could not be explained by simple 2-centre-2-electon interactions and that a delocalized model was required. The critical breakthroughs in rationalization of their structures occurred in the early 1970s. First, Williams recognized that the structures of the boranes fell into well-defined groups, being either closed polyhedra (given the prefix *closo*) or open fragments thereof (*nido*, *arachno*, *hypho*, etc.) [4]. This was important because previously there had been the general belief that structurally the majority of the open boranes were simply fragments of an icosahedron. Williams identified that, e.g., the structure of B_10_H_14_ was in fact a *nido* fragment of the 11-vertex octadecahedron, whilst that of B_5_H_11_ was an *arachno* fragment of the 7-vertex pentagonal bipyramid. These structural relationships between the closed polyhedra and their *nido* and *arachno* fragments were later to be popularized in a classic figure by Rudolph [5], Figure 1. Second, and most importantly, Wade subsequently provided an explanation for these structural patterns in terms of the number of skeletal electron pairs (SEPs) available [6]. In brief, for an *n* vertex polyhedron a *closo* species has (*n* + 1) SEPs, a *nido* species (*n* + 2), an *arachno* species (*n* + 3), etc.

A particularly important feature of boron cluster chemistry is the number and diversity of heteroatom fragments, from across a broad spectrum of the periodic table, that can occupy vertices of the polyhedron. Thus, for example, in [3-(C_5_H_5_)-*closo*-3,1,2-CoC_2_B_9_H_11_] [7,8] the Co and C atoms occupy vertices of the resulting icosahedron (Figure 2a), whilst in [*nido*-6-SB_9_H_11_] [9] the S atom occupies a vertex of the resulting *nido* fragment of an octadecahedron (Figure 2b). Importantly, the same electron-counting rules which describe the boranes apply to these heteroboranes, with the {CoCp} fragment providing 1 vertex and 2 skeletal electrons, the {CH} fragments 1 vertex and 3 skeletal electrons and the {S} fragment 1 vertex and 4 skeletal electrons.

In this contribution we are concerned with heteroboranes in which a main group heteroatom *could* either be considered as a vertex or as a bridge to the cluster, and we describe a method to distinguish between these alternatives based on the differing effects that a heteroatom bridge or vertex has on the structure of the {B*_x_*} fragment of the molecule. To illustrate the problem, consider the species [Me_3_NCB_10_H_10_PPh] [10], assigned the refcode PACBOR in the Cambridge Structural Database (CSD) [11]. As described in the publication, and as implied by Figure 3a, PACBOR is considered to be an 11-vertex *nido* CB_10_ species with a {PPh} fragment bridging a B—B edge in the open face. In this description the P atom is not formally a vertex, the {PPh} unit contributes just 2e to the cluster bonding, and consequently the {B_10_} residue would be an *arachno* fragment of an icosahedron, characterized by 13 SEP. Alternatively, simply redrawing the molecule with two additional P—B connectivities (Figure 3b) implies that the P atom is a cluster vertex. In this description PACBOR would be a 12-vertex *nido* PCB_10_ species in which the {PPh} fragment contributes 1 vertex and 4 skeletal electrons. Now the {B_10_} residue would be described as a *hypho* fragment of a 13-vertex parent (docosahedron), characterized by 14 SEP.

To distinguish between these alternatives, and hence to afford the more appropriate description of PACBOR, we need to see if the experimental {B_10_} residue fits better with exemplar *arachno* or *hypho* {B_10_} fragments, but first we need to establish if such exemplar fragments are themselves distinguishable. This is because, as shown in Figure 4, these fragments are topologically identical and, moreover, they are further topologically identical with a 12-SEP *nido* {B_10_} fragment. So, faced with an experimental {B_10_} fragment of this topology (from a crystallographic study) how do we know if it is best described as *nido*, *arachno* or *hypho*? More generally, how do we distinguish between topologically equivalent {B*_x_*} fragments of different parent polyhedra? One approach for the {B_10_} fragments could be to use the dimensions of the fragment, since as we go from *nido* {B_10_} to *arachno* {B_10_} to *hypho* {B_10_} the cluster gets progressively shorter, deeper and broader (see Appendix A for detail) but this is not the best method and certainly not general for other fragments. Rather we make use of *Structure Overlay* (SO) calculations implemented in Mercury [12], as described in the Method section. The present study complements our earlier one using the same principle of structure overlay to distinguish between transition- or post-transition-metal bridges and vertices in species of the type [*M*B_10_H_12_] [13].

## 2. Results

### 2.1. PACBOR, [Me_3_NCB_10_H_10_PPh]

As noted above, for PACBOR the possible alternatives are as follows; if the P atom is merely a bridge the {B_10_} residue is an *arachno* fragment of an icosahedron but if the P atom is a vertex the {B_10_} residue is a *hypho* fragment of a docosahedron. These exemplar fragments with the same topology were taken from DFT-optimized [B_12_H_12_]^2−^ and [B_13_H_13_]^2−^, respectively and a structure overlay calculation performed between them. See Table 1 for atom numbering and SO results. A rms misfit value of 0.109 Å confirms that the exemplar fragments can be distinguished. The {B_10_} residue of PACBOR, taken from the crystallographic study, was then overlaid against each of the exemplar fragments affording the results shown in Table 2. A rms misfit of 0.053 Å for the *hypho* fragment against a rms misfit of 0.083 Å for the *arachno* fragment shows that the {B_10_} residue of PACBOR is better described as *hypho*. Consequently, the P atom is better regarded as a vertex than as a bridge meaning that the molecule overall is better described as a *nido* 12-vertex phosphacarborane than a *nido* 11-vertex carborane with a μ-PPh. Note that this conclusion is at variance with the description of PACBOR in both the original publication [10] and the name of this compound assigned in the CSD.

In an attempt to describe concisely the nature of the P atom in PACBOR (i.e., bridge or vertex) we re-introduce [13] the term verticity. For an atom X bonded to an experimental {B*_x_*} fragment A overlaid against two exemplar {B*_x_*} fragments B and C (with C being the more open fragment):Verticity of X = {(rms misfit B/C + rms misfit A/B − rms misfit A/C)/2 × (rms misfit B/C)} × 100%

This affords verticity on a scale from 0% (misfit of A with B is 0) to 100% (misfit of A with C is 0). For PACBOR the verticity of the P atom is thus calculated as:Verticity of P = {(0.109 + 0.083 − 0.053)/2 × 0.109} × 100%
i.e., a verticity of 63.8%. Clearly, a verticity > 50% means that the atom should be regarded more as a vertex than as a bridge whereas a verticity < 50% implies the opposite.

### 2.2. The Thermodynamic Isomer of [C_2_B_10_H_13_]^−^

Two-electron reduction of either [*closo*-1,2-C_2_B_10_H_12_] or [*closo*-1,7-C_2_B_10_H_12_] affords [*nido*-7,9-C_2_B_10_H_12_]^2−^, whose structure is that of a fragment of a docosahedron. Protonation of the dianion yields [C_2_B_10_H_13_]^−^ [14], known in two isomeric forms, a kinetic isomer which retains the structure of the dianion and a thermodynamic isomer, usually described as comprising a *nido* 11-vertex CB_10_ cage (fragment of an icosahedron) whose open face has a {CH_2_} group bridging a B—B edge. Although neither the kinetic nor thermodynamic isomers per se have been studied crystallographically they are both so characterized in derivative form, the kinetic isomer as the [PMePh_3_]^+^ salt of the *C*,*C*’-dimethyl analogue [15]. There are four crystallographic determinations of derivatives of the thermodynamic isomer [16,17,18], of which two, MMUHDB [17] and YACRIE [18], are of high precision. We have studied MMUHDB and YACRIE to establish if the ‘bridging’ C atom is better regarded as a bridge or a vertex.

The anion MMUHDB and molecule YACRIE are shown in Figure 5. The ‘bridging’ unit is {C(H)Me} with the Me group *exo* and the C—H bond *endo* to the polyhedron. The alternative descriptions of the {B_10_} fragment here are the same as for PACBOR; if the {C(H)Me} unit is a bridge the polyhedron is a *nido* 11-vertex CB_10_ cage of which the {B_10_} fragment would be described as *arachno*, but if the ‘bridging’ C is really a vertex we have a *nido* 12-vertex C_2_B_10_ system of which the {B_10_} fragment would be *hypho*. In Table 3 and Table 4 are the results of SO calculations between the experimental MMUHDB and YACRIE {B_10_} fragments and the exemplar fragments. In both cases the better fit is with the *hypho* exemplar, implying that the ‘bridging’ C atom is better described as a cluster vertex rather than a bridge. Calculated verticities for the C atom are 60.1% and 61.0%, respectively.

Interestingly, although MMUHDB is described as having a bridging {C(H)Me} unit [17] the species is named in the CSD as though both C atoms are vertices. Conversely, the publication describing YACRIE perceptively suggests that the ‘bridging’ C be regarded ‘*as an integral part of the cluster*’ [18] whilst in the CSD the compound is named as having a μ-(9,10-ethylidene) fragment.

### 2.3. ZONCOU and FAFYAN, a Case of Mistaken Identity

In 1995 Paetzold et al. published the synthesis and structure of ZONCOU, [PhNB_11_H_11_NEt_3_], Figure 6a [19], and in 2002 the same group reported FAFYAN, [MeNB_11_H_11_NHEt_2_], Figure 6b [20]. Although the Paetzold group described both species as azaboranes, thus clearly identifying the N(Ph) and N(Me) atoms as vertices, in the CSD FAFYAN is named as a 12-vertex azaborane whilst ZONCOU is described as an 11-vertex borane with a triply bridging phenylimino group.

This has prompted us to examine these two structures by SO calculations to establish the true nature of the N(Ph) and N(Me) atoms. The first obvious comment is that the two cages ‘look’ almost superimposable, implying that one is incorrectly described in the CSD. The second important point is that the N atoms in question make *three* short connectivities to B atoms, 1.5–1.6 Å. This is quite different to the situation in the previous species, PACBOR, MMUHDB and YACRIE, where the P or C atoms made only *two* short connections, ca. 2.0 Å in PACBOR and ca. 1.6 Å in MMUHDB and YACRIE, and implies that the N(Ph) and N(Me) atoms in ZONCOU and FAFYAN have considerable vertex character.

This is confirmed by our analysis. Initially we overlaid the {B_11_} fragments of ZONCOU and FAFYAN with each other, with the results shown in Table 5. The extremely small rms misfit value, 0.019 Å, confirms that the two cages are isostructural. We next compared each of the experimental {B_11_} fragments with exemplars, but now we need different exemplars to those used previously. If the N atom is a bridge the {B_11_} unit is a *nido* fragment of an icosahedron, but if it is a vertex the cage is a *nido* NB_11_ cluster the {B_11_} component of which is an *arachno* fragment of a docosahedron. Note that in this description the *nido* 12-vertex NB_11_ parent has a different topology (a degree-5 vertex missing) to the *nido* 12-vertex PCB_10_ parent of PACBOR and *nido* C_2_B_10_ parent of MMUHDB and YACRIE (the degree-4 vertex missing). These differing topologies are possible because of the relatively low symmetry (*C*_2v_) of the docosahedron.

The result of overlaying the exemplars is shown in Table 6. The rms misfit value is 0.170 Å, in large measure due to a substantial misfit of B8, meaning that the two exemplars can be distinguished structurally. Next, we overlaid the {B_11_} fragments of ZONCOU and FAFYAN with each of the exemplar {B_11_} fragments. Because ZONCOU and FAFYAN are isostructural we present only the results for ZONCOU (Table 7) with those for FAFYAN deposited as Appendix A.

Clearly there is a much better fit of the experimental {B_11_} fragment with the *arachno* exemplar (again note the very large misfit of B8 with the *nido* exemplar) meaning that the N(Ph) atom in ZONCOU should be regarded as a vertex rather than as a bridge. The verticity of N(Ph) is calculated as 92.3%. For FAFYAN the results are very similar; rms misfits of 0.227 Å against the *nido* exemplar and 0.081 Å against the *arachno* exemplar, resulting in a verticity of N(Me) of 92.9%.

In summary, both ZONCOU and FAFYAN are clearly azaboranes, with the former erroneously described in the CSD. The extremely high verticities are fully consistent with the N vertices making three short connections to B atoms in the cluster.

### 2.4. YELXES, S Bridge or Vertex?

The molecule [MeS_2_B_9_H_10_], YELXES [21], is shown in perspective view in Figure 7. It features two heteroatoms one, S2, making three short connections to B atoms and the other a S(Me) unit bound to only two B atoms. As noted in the previous section the S2 heteroatom connected to three B atoms can be reasonably assumed to be a cluster vertex, and the question that then arises is whether the S(Me) atom is better regarded as a bridge or as a vertex.

If the S(Me) atom is a true vertex the cluster in YELXES is an *arachno* {S_2_B_9_} polyhedron, a fragment of a docosahedron although with a different topology to the *arachno* {B_11_} fragments of ZONCOU and FAFYAN, again a consequence of the low symmetry of the docosahedron. Although this description is favored in the publication describing YELXES [21], *two* alternatives for a S(Me) bridge are also discussed, that in which it is a 3-e 2-orbital donor and that in which it is a 1-e 1-orbital source. If the former the cluster would be *arachno* {SB_9_}, a fragment of an icosahedron with the same topology as the *arachno* {B_10_} fragments of PACBOR, MMUHDB and YACRIE. If the latter the cluster would be *nido* {SB_9_}, derived from an octadecahedron by removal of the degree-6 vertex.

This leads to three possible descriptions of the {B_9_} fragment of YELXES; the {B_9_} fragment of an *arachno* {S_2_B_9_} cluster would be a *klado* fragment of a docosahedron [22], the {B_9_} fragment of an *arachno* {SB_9_} cluster would be a *hypho* fragment of an icosahedron; and the {B_9_} fragment of a *nido* {SB_9_} cluster would be an *arachno* fragment of an octadecahedron. To illustrate these options Figure 8 shows (a) a numbered docosahedron, (b) a *klado* fragment thereof, (c) the molecule YELXES reoriented to present the same view of the {B_9_} fragment, (d) an icosahedron, (e) a *hypho* fragment thereof, (f) an octadecahedron, and (g) an *arachno* fragment thereof. The three {B_9_} fragments all share the same topology and numbering.

Thus, there are now three exemplar {B_9_} fragments to consider, *arachno*, *hypho* and *klado* and in Table 8 are presented the results of SO calculations in which the three exemplar fragments are compared. The rms misfit values confirm that all three possibilities may be structurally distinguished, and we note that the two largest misfits are between *klado*/*hypho* and *klado*/*arachno*, i.e., between situations where the S(Me) is a vertex and both the bridge options.

Table 9 shows the result of SO calculations of the {B_9_} fragment of YELXES with each of these exemplars. The fit is best with the *klado* exemplar suggesting that YELXES is best regarded as an *arachno* S_2_B_9_ cluster in which the S(Me) atom is a vertex, rather than an *arachno* SB_9_ cluster with a 3-e 2-orbital bridging S(Me) or a *nido* SB_9_ cluster with a 1-e 1-orbital bridging S(Me). Note that a similar bonding mode for the two S(Me) units of [(MeS)_2_B_6_H_8_] has previously been suggested [23]. In YELXES the verticity of the S(Me) atom is 62.3% using the *klado* and *hypho* {B_9_} misfit values, and 59.5% using the *klado* and *arachno* values.

### 2.5. BUPPEI, an Example Too Far?

In the preceding sections we have shown that the ‘bridging’ atoms (which come from groups 14, 15 and 16) in PACBOR, MMUHDB, YACRIE and YELXES are better described as vertices rather than bridges, and to extend our study we were therefore interested in an example involving a group 13 element. The anion BUPPEI (Figure 9a) consists of a {B(H)NMe_2_} unit bonded to an open 13-vertex C_2_B_11_ cage via two short B—B connectivities [24]. The same publication also describes the anion BUPPIM (Figure 9b) which has an analogous structure but with only a bridging H atom. The bridging H in BUPPIM can only be a bridge, but in principle the {B(H)NMe_2_} unit in BUPPEI could be either a bridge or vertex, so we have analyzed these two structures by SO calculations.

If the ‘bridging’ B in BUPPEI is merely a bridge the cluster is a *nido*-C_2_B_11_ one of which the {B_11_} unit would be a *hypho* fragment of the 14-vertex parent [B_14_H_14_]^2−^, but if it is a vertex the cage is *nido*-C_2_B_12_ and the {B_11_} fragment would be a *klado* fragment of the 15-vertex [B_15_H_15_]^2−^. For BUPPIM the only option is that the {B_11_} fragment is *hypho*. In Table 10 we show the result of a SO calculation between these two exemplar {B_11_} fragments. The individual atom misfits are greatest at the B atoms in the open face (B1, B2, B3, B4) and at B10, the atom below the cage C atoms. The large rms misfit of 0.201 Å establishes that the exemplars can be distinguished.

In Table 11 we overlay the experimental {B_11_} fragment of BUPPEI (unprimed cage, one of two crystallographically-independent anions) with the exemplar *hypho* and *klado* fragments sharing the same topology. Clearly the fit is better with the less open, *hypho* fragment (rms misfit 0.062 Å) than with the more open *klado* fragment (rms misfit 0.151 Å). This affords a verticity for the ‘bridging’ B of only 27.9%, implying it really is better described as a bridge and not a vertex. Appendix A contains the almost identical results of the equivalent calculations using the primed cage; rms misfit vs. *hypho* exemplar 0.063 Å, vs. *klado* exemplar 0.147 Å, verticity of B 29.1%.

For comparison we have also overlaid the {B_11_} fragment of BUPPIM with the *hypho* and *klado* exemplars, and the results are shown in Table 12. The rms misfit vs. the *hypho* exemplar is 0.047 Å (smaller than that for BUPPEI) and that vs. the *klado* exemplar is 0.181 Å (larger than that for BUPPEI). This affords a ‘verticity’ for the bridging H atom (which, of course, can only be a bridge) of 16.7%. Although these structural differences between the {B_11_} fragments of BUPPEI and BUPPIM are only small we believe they are important, as detailed in the ‘Discussion’ section below.

## 3. Discussion

We have used *Structure Overlay* calculations to analyze boron clusters which contain an open face onto which a main group heteroatom *X* is bound by two short B—*X* connectivities for a variety of atoms *X*; B (BUPPEI [24]), C (MMUHDB [17] and YACRIE [18]), P (PACBOR [10]) and S (YELXES [21]). Our approach is based on the fact that considering the atom *X* as a bridge or a cluster vertex will result in different formal descriptions of the {B*_x_*} residue of the cluster. We have overlaid the structure of the experimental {B*_x_*} residue with those of exemplar {B*_x_*} fragments having the same topology derived from *closo* species [B*_n_*H*_n_*]^2−^ optimized by DFT calculation, and used the better or best fit to identify the better or best description for the experimental {B*_x_*} residue and hence better or best description of *X* as a bridge or vertex.

In describing their experimental results, some authors have simply assumed that *X* is a bridge and not a formal part of the cluster (PACBOR, MMUHDB and YACRIE, although in the last case the authors also cast some doubt about this assumption). Others have argued (as it turns out correctly) that *X* is a vertex whilst also discussing the possibility of it as a bridge (YELXES). Yet, others have assumed (as it turns out incorrectly) that *X* is a vertex without any explanation (BUPPEI). However, as far as we are aware no previous method to distinguish between bridge or vertex options has been proposed.

We find that, with the exception of BUPPEI, in all the cases we have considered *X* is better described as a vertex and we have attempted to quantify this with the term *verticity*. For MMUHDB, YACRIE, PACBOR and YELXES the calculated verticities are remarkably consistent, 60–65%. We recognize that these values are not particularly high, and certainly not as high as those (>90%) we have found when similarly analyzing heteroatoms bound by three short B connectivities (ZONCOU [19] and FAFYAN [20]) [25]. Nevertheless, verticities of ca. 60% do suggest that the better description of the atom *X* in MMUHDB, YACRIE, PACBOR and YELXES is as a cluster vertex. The exception is BUPPEI, in which the verticity of the bridging B atom is only ca. 28%, implying that this atom is better described as a bridge than a vertex, and that the correct formal description of the anion is as a *nido* C_2_B_11_ species with a bridging B and not as a *nido* C_2_B_12_ species [24]. How can we rationalize these results?

Formally, electron donation to the cluster from all these ‘bridging’ units is from two sources; those electrons involved in the (radial) interaction between *X* and the cluster (i.e., in simple terms those electrons donated via the B—*X* connectivities) and those electrons in either a (tangential) lone pair or 2-centre-2-electron bond on *X* which is *endo* with respect to the cluster. The {C(H)Me} fragments of MMUHDB and YACRIE provide 2 radial electrons as does the {PPh} fragment of PACBOR, whilst the {SMe} group of YELXES provides 3 such electrons. In BUPPEI, however, only 1 radial electron is formally donated to the cluster. In all cases these contributions to the overall skeletal electron count are then formally enhanced by the *endo* electron pair. Note that in ZONCOU and FAFYAN the orientation of the Ph or Me substituent implies no *endo* lone pair on the N atoms, with all the (4) donated electrons being radial.

Our structure overlay results, and the conclusions regarding the formal descriptions of *X* as either a bridge or vertex which derive from them, may be understood at a simple level if we assume that the radial electron donation from *X* is more important than donation of the *endo* electron pair. In this way the 2- or 3-electron radial donations of {C(H)Me}, {PPh} and {SMe}, supplemented by relatively weak *endo* donation, achieve verticities > 50% (making the atom *X* formally a vertex) but they do not approach the very high verticities seen in ZONCOU and FAFYAN where the N heteroatoms are fully integrated into the cluster framework and the total skeletal electron count does not depend on *endo* electrons. In contrast weak *endo* donation is insufficient to raise the verticity in BUPPEI (which has only 1-electron radial donation) above the 50% threshold and consequently the {B(H)NMe_2_} group is formally described as a bridge. Note, however, that the {B(H)NMe_2_} in BUPPEI has a higher verticity than the simple bridging H atom in BUPPIM since in the latter case there are no *endo* electrons to be donated.

A useful test of our explanation would be consider the effect of a ‘bridging’ {B(H)NMe_3_} unit as opposed to a bridging {B(H)NMe_2_} unit, since the former would provide 2 radial electrons as opposed to just 1 and should, therefore, assume more vertex character. Since we have been unable to locate a (hetero)borane with a ‘bridging’ {B(H)NMe_3_} fragment in the CSD we have constructed one from BUPPEI. Initially (to check the validity of the process) we have simply optimized the anion BUPPEI by DFT calculation and in Appendix A are presented the results of structure overlay calculations of the {B_11_} fragment of the optimized model and the unprimed and primed experimental {B_11_} fragments. In both cases the very small rms misfit values of 0.019 Å confirm the validity of the optimization. Consequently, we have constructed the neutral species [(CH_2_)_3_C_2_B_11_H_11_{B(H)NMe_3_}] (compound **1**) from the coordinates of BUPPEI and optimized its structure. Figure 10b shows a perspective view of **1** (since this is an optimized model and not a real molecule B atoms are shown in pink) adjacent to BUPPEI (Figure 10a). Note that, in terms of their overall skeletal electron count, the anion BUPPEI and neutral species **1** are isoelectronic, the anionic charge of BUPPEI being compensated by an additional radial electron from the bridge in **1**. Initial comparison of the two structures does suggest a somewhat greater degree of vertex character for the bridging B of **1** since, in addition to the two short B—B distances of 1.788 and 1.790 Å (c.f. 1.868–1.890 Å in BUPPEI) the bridging B is only 2.155 and 2.168 Å from the other two B atoms in the open face (c.f. 2.388–2.442 Å in BUPPEI).

Table 13 reports the results of structure overlay calculations of the {B_11_} fragment of **1** with both the *hypho* and *klado* {B_11_} exemplars. Although the rms misfit with the *hypho* exemplar is still the smaller (meaning that the bridging B is still better described as a bridge) it is greater than that for BUPPEI and, correspondingly, the rms misfit with the *klado* exemplar is smaller than that for BUPPEI. For the bridging B atom the verticity is 44.0% in compound **1** compared to only ca. 28% in BUPPEI, supporting the idea that formally increasing the radial electron donation of the bridging unit from 1 to 2 electrons significantly increases the vertex character.

## 4. Conclusions

A number of borane and heteroborane compounds are known in which main group atoms ‘bridge’ a B—B connectivity on the open face and a valid question is whether that ‘bridge’ really is just a bridge or if it is better regarded as a vertex in a larger cluster. We have used *Structure Overlay* calculations comparing the {B*_x_*} fragments of crystallographically studied species with those of exemplar fragments taken from optimized alternates in an attempt to address this question. We find that in most cases the main group ‘bridge’ is, in fact, better regarded as a vertex. We have rationalized our results by considering that the radial electron donation of a ‘bridging’ unit makes a more significant contribution to the overall skeletal electron count than does donation of an *endo* electron pair.

In an earlier study [13] we have used structure overlay calculations to analyze the bridge or vertex nature of transition-metal and post-transition-metal fragments in the family of metallaboranes [*M*B_10_H_12_]. The essential finding of this work was that a strong correlation existed between verticity and the number of available frontier orbitals of the metal fragment. Overall, there are two important conclusions common to both studies: First, the principle of structure overlay is highly appropriate in comparing the structures of two polyhedra since there is generally an unmanageable number of possible parameters (e.g., bond lengths, interbond angle, torsion angles etc) for such conventional comparison. Second, the verticity of a fragment, be it main group or transition-metal, in a heteroborane can span a wide range of values. It is not always the case that the fragment is very obviously a bridge or very obviously a vertex and a pragmatic approach to the question of bridge or vertex has to be adopted; nevertheless, via structure overlay calculations, we can now at least conclude that, for example, ‘the atom is better regarded as a vertex than a bridge.’

## 5. Method

As noted in the Introduction, whether a heteroatom is a bridge on or a vertex in a boron-based cluster results in a difference of 1 atom in the vertex count and 1 SEP in the skeletal electron count of the whole cluster and hence the {B*_x_*} residue. The two alternative {B*_x_*} fragments are topologically equivalent. The experimental {B*_x_*} fragment is taken from a crystallographic study of the heteroborane. Wherever possible we restrict ourselves to studies of high precision which are free from crystallographic disorder, at least in the cluster fragment. We then construct appropriate exemplar {B*_x_*} fragments by optimizing by DFT calculation (BP86/6-31G**; see Appendix A for details) the *closo* species [B*_n_*H*_n_*]^2−^ and then excluding vertices to afford the {B*_x_*} fragment with correct topology. Next, we perform a structure overlay calculation on the two alternative exemplar fragments. Output from this is a list of individual atom misfit values and an overall rms misfit (Å), a high rms misfit showing that the two exemplar fragments can be distinguished. We then overlay the experimental {B*_x_*} fragment with each of the exemplar {B*_x_*} fragments to establish which is the better fit (lower rms misfit). This, then, provides evidence for describing the heteroatom as either a bridge or vertex.

## Figures and Tables

**Figure 1 molecules-28-00190-f001:**
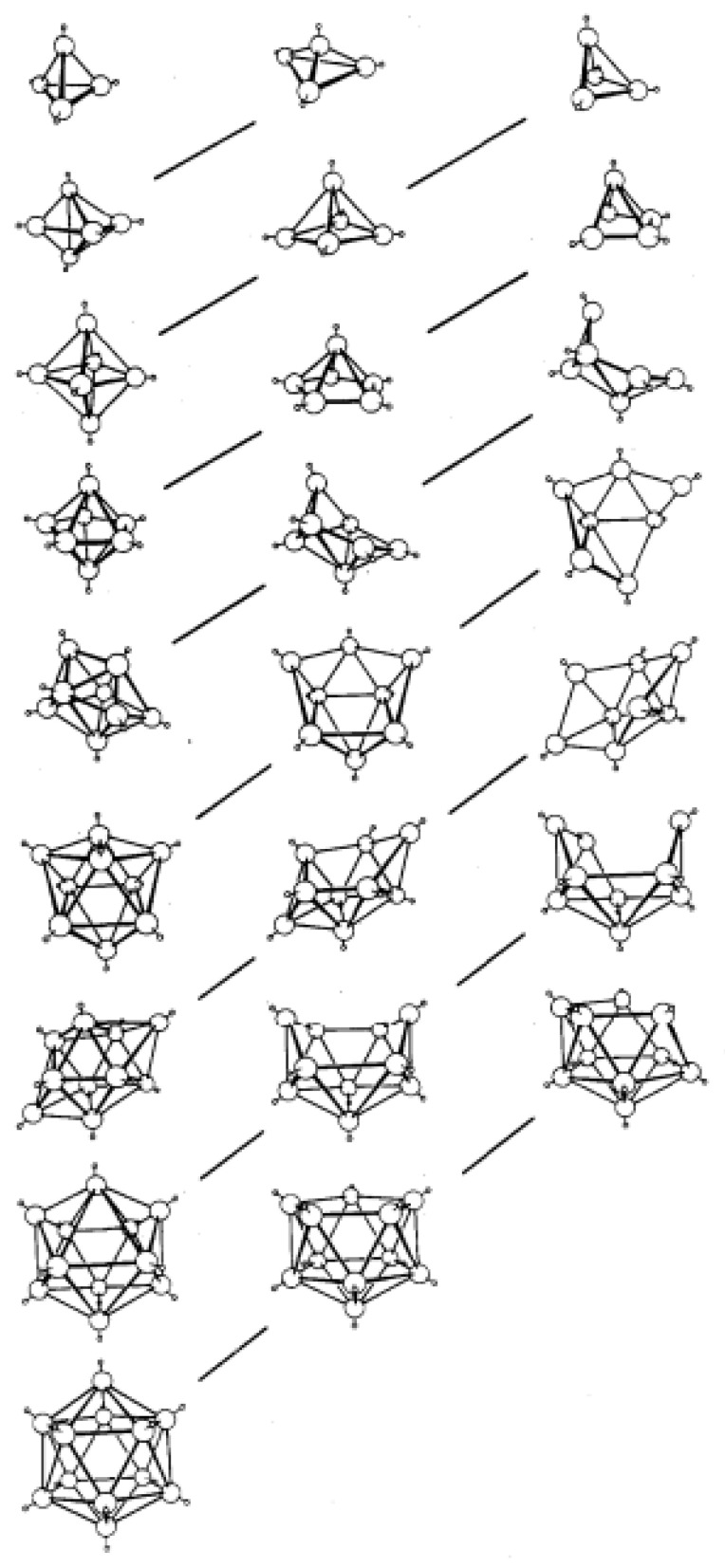
Idealized *closo* (left column), *nido* (middle column) and *arachno* polyhedra composed of {BH} fragments with the structural relationships between them illustrated by diagonal lines. Reproduced with permission from reference [5].

**Figure 2 molecules-28-00190-f002:**
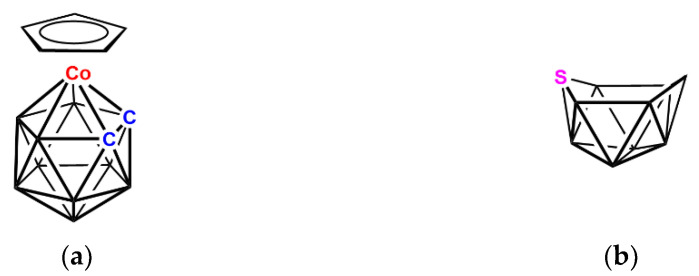
Line diagrams of (**a**) the metallacarborane [3-(C_5_H_5_)-*closo*-3,1,2-CoC_2_B_9_H_11_] and (**b**) the thiaborane [*nido*-6-SB_9_H_11_]. Unlabeled vertices are {BH} fragments. In (**a**) the vertices C are {CH} fragments and in (**b**) there are additionally two H bridges on the open face (not shown).

**Figure 3 molecules-28-00190-f003:**
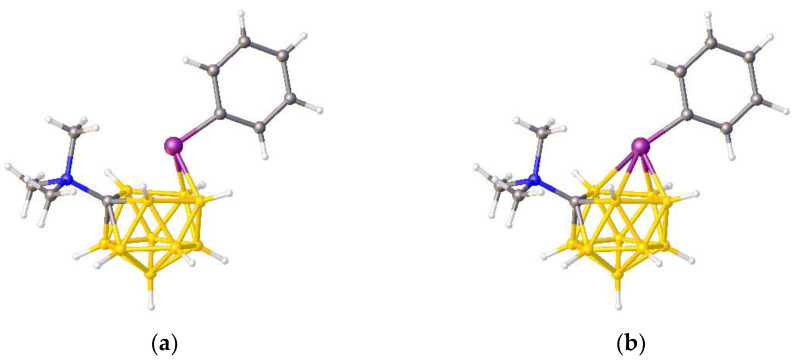
The molecule PACBOR. In (**a**) the P atom (purple) is drawn with only two connectivities to B atoms (yellow) whereas in (**b**) there are four P—B connectivities. N atom in blue and C atoms in grey.

**Figure 4 molecules-28-00190-f004:**
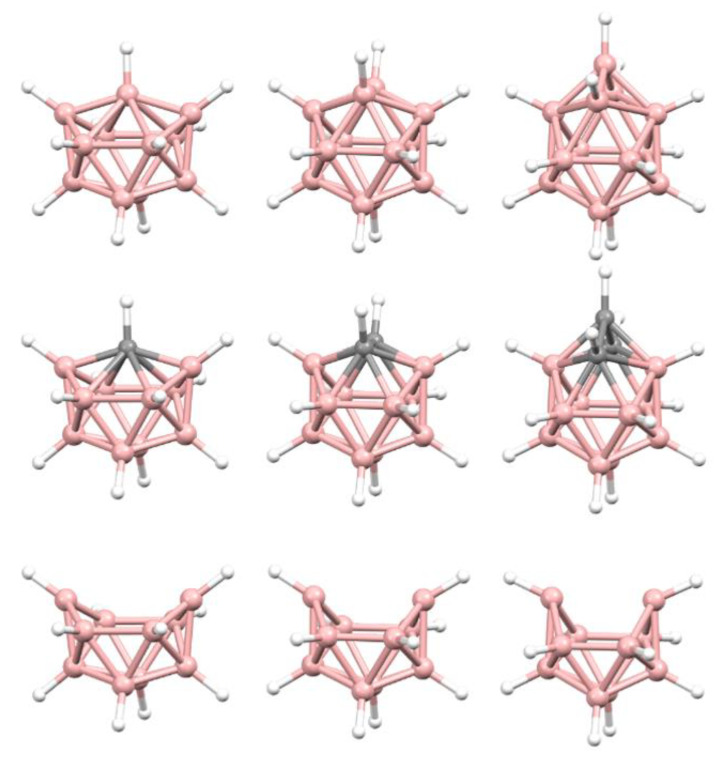
Top row, left to right; the 11-, 12- and 13-vertex boranes [B_11_H_11_]^2−^, [B_12_H_12_]^2−^ and [B_13_H_13_]^2−^. Middle row, left to right; 1, 2 and 3 atoms (grey) are identified for removal. Bottom row, left to right; the 10-vertex *nido*, *arachno* and *hypho* fragments which remain. All the polyhedra in the bottom row are topologically equivalent.

**Figure 5 molecules-28-00190-f005:**
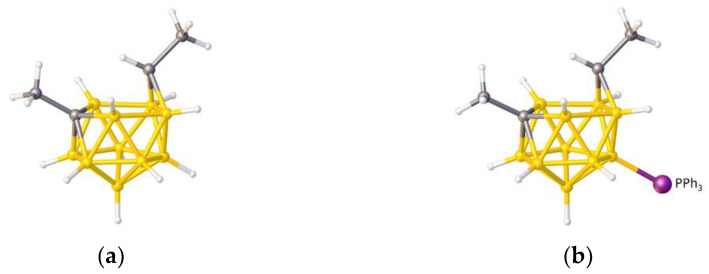
(**a**) The anion MMUHDB. (**b**) The molecule YACRIE. B atoms in yellow, C atoms in grey and P (YACRIE) in purple.

**Figure 6 molecules-28-00190-f006:**
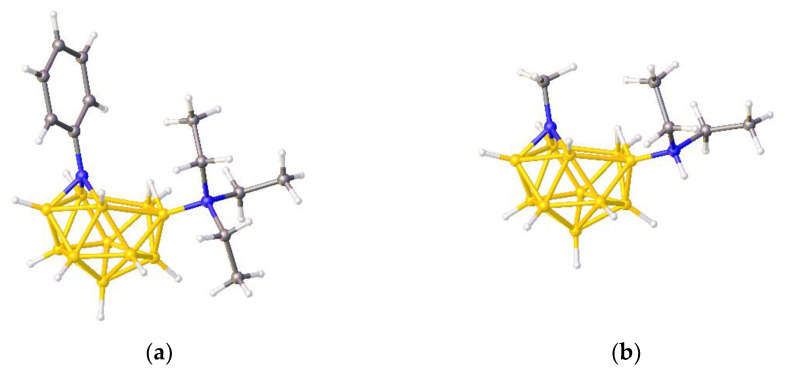
(**a**) The molecule ZONCOU. (**b**) The molecule FAFYAN. B atoms in yellow, C atoms in grey and N in blue.

**Figure 7 molecules-28-00190-f007:**
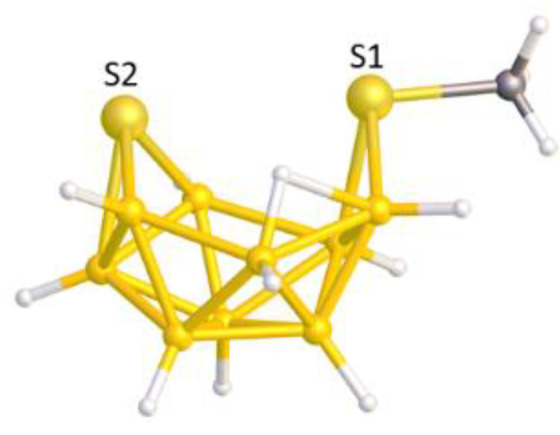
The molecule YELXES. B and S atoms in yellow, C atom.

**Figure 8 molecules-28-00190-f008:**
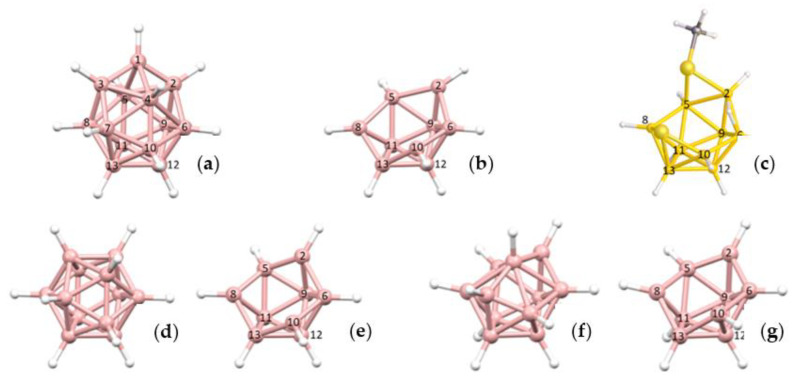
(**a**) The docosahedron; (**b**) A *klado* fragment thereof; (**c**) YELXES (B and S atoms in yellow, C atoms in grey) in the same orientation; (**d**) The icosahedron; (**e**) A *hypho* fragment thereof; (**f**) The octadecahedron; (**g**) An *arachno* fragment thereof.

**Figure 9 molecules-28-00190-f009:**
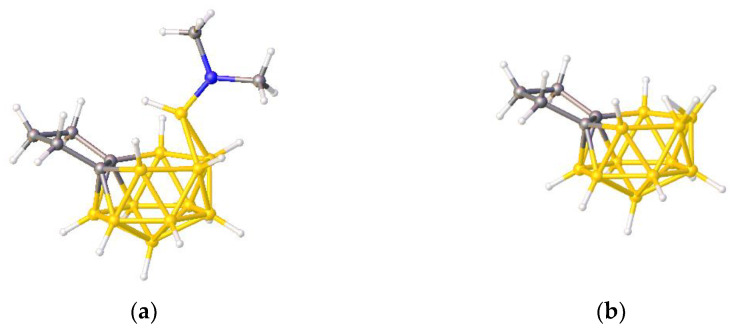
(**a**) The anion BUPPEI. (**b**) The anion BUPPIM. B atoms in yellow, C atoms in grey and N in blue.

**Figure 10 molecules-28-00190-f010:**
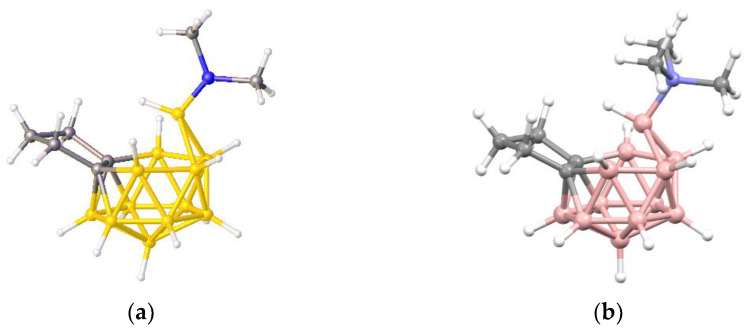
(**a**) The anion BUPPEI (B atoms in yellow). (**b**) The computed structure of **1** (B atoms in pink). C atoms in grey and N in blue for both species.

**Table 1 molecules-28-00190-t001:** Structure Overlay calculation between the exemplar *arachno* {B_10_} fragment (left) and exemplar *hypho* {B_10_} fragment (right).

	Atom	Misfit (Å)	
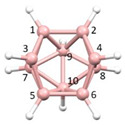	1	0.111	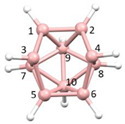
	2	0.111	
	3	0.163	
	4	0.163	
	5	0.111	
	6	0.111	
	7	0.086	
	8	0.086	
	9	0.028	
	10	0.028	
	**rms**	**0.109**	

**Table 2 molecules-28-00190-t002:** Structure Overlay calculations (Å) between the experimental {B_10_} fragment of PACBOR and exemplar *arachno* {B_10_} fragment (left) and exemplar *hypho* {B_10_} fragment (right).

	PACBOR/*a*-{B_10_}			PACBOR/*h*-{B_10_}		
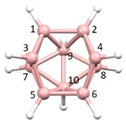	1	0.088	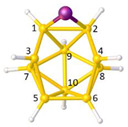	1	0.047	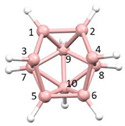
	2	0.083		2	0.045	
	3	0.127		3	0.089	
	4	0.131		4	0.086	
	5	0.075		5	0.044	
	6	0.070		6	0.042	
	7	0.062		7	0.025	
	8	0.076		8	0.014	
	9	0.013		9	0.030	
	10	0.035		10	0.050	
	**rms**	**0.083**		**rms**	**0.053**	

**Table 3 molecules-28-00190-t003:** Structure Overlay calculations (Å) between the experimental {B_10_} fragment of MMUHDB and exemplar *arachno* {B_10_} fragment (left) and exemplar *hypho* {B_10_} fragment (right).

	MMUHDB/*a*-{B_10_}			MMUHDB/*h*-{B_10_}		
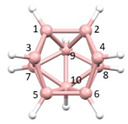	1	0.102	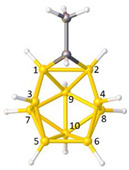	1	0.060	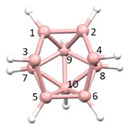
	2	0.091		2	0.059	
	3	0.151		3	0.128	
	4	0.150		4	0.132	
	5	0.061		5	0.061	
	6	0.062		6	0.059	
	7	0.083		7	0.014	
	8	0.084		8	0.014	
	9	0.018		9	0.032	
	10	0.020		10	0.026	
	**rms**	**0.093**		**rms**	0.071	

**Table 4 molecules-28-00190-t004:** Structure Overlay calculations (Å) between the experimental {B_10_} fragment of YACRIE and exemplar *arachno* {B_10_} fragment (left) and exemplar *hypho* {B_10_} fragment (right).

	YACRIE/*a*-{B_10_}			YACRIE/*h*-{B_10_}		
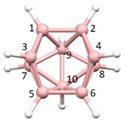	1	0.111	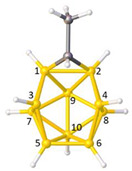	1	0.076	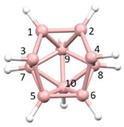
	2	0.105		2	0.075	
	3	0.151		3	0.122	
	4	0.157		4	0.126	
	5	0.067		5	0.057	
	6	0.064		6	0.055	
	7	0.074		7	0.021	
	8	0.077		8	0.017	
	9	0.030		9	0.046	
	10	0.024		10	0.011	
	**rms**	**0.096**		**rms**	**0.072**	

**Table 5 molecules-28-00190-t005:** Structure Overlay calculation between the {B_11_} fragments of ZONCOU (left) and FAFYAN (right).

	Atom	Misfit (Å)	
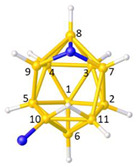	1	0.008	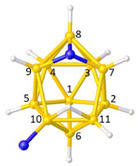
	2	0.018	
	3	0.009	
	4	0.010	
	5	0.009	
	6	0.017	
	7	0.021	
	8	0.017	
	9	0.027	
	10	0.026	
	11	0.026	
	**rms**	**0.019**	

**Table 6 molecules-28-00190-t006:** Structure Overlay calculation between the exemplar *nido* {B_11_} fragment (left) and exemplar *arachno* {B_11_} fragment (right).

	Atom	Misfit (Å)	
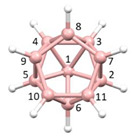	1	0.001	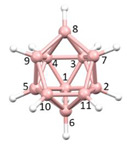
	2	0.084	
	3	0.045	
	4	0.045	
	5	0.083	
	6	0.067	
	7	0.181	
	8	0.400	
	9	0.181	
	10	0.188	
	11	0.188	
	**rms**	**0.170**	

**Table 7 molecules-28-00190-t007:** Structure Overlay calculations (Å) between the experimental {B_11_} fragment of ZONCOU and exemplar *nido* {B_11_} fragment (left) and exemplar *arachno* {B_11_} fragment (right).

	ZONCOU/*n*-{B_11_}			ZONCOU/*a*-{B_11_}		
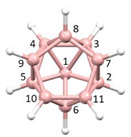	1	0.052	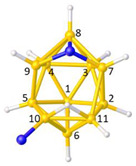	1	0.051	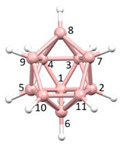
	2	0.174		2	0.093	
	3	0.030		3	0.026	
	4	0.031		4	0.030	
	5	0.167		5	0.087	
	6	0.105		6	0.043	
	7	0.203		7	0.095	
	8	0.497		8	0.095	
	9	0.210		9	0.121	
	10	0.338		10	0.152	
	11	0.323		11	0.140	
	**rms**	**0.238**		**rms**	**0.094**	

**Table 8 molecules-28-00190-t008:** Structure Overlay calculations (Å) between exemplar *klado*, *hypho* and *arachno* {B_9_} fragments.

Vertex	*k*-{B_9_}/*h*-{B_9_}	*k*-{B_9_}/*a*-{B_9_}	*h*-{B_9_}/*a*-{B_9_}
2	0.338	0.205	0.185
5	0.165	0.168	0.143
6	0.043	0.127	0.142
8	0.240	0.247	0.137
9	0.035	0.080	0.070
10	0.114	0.096	0.137
11	0.057	0.017	0.045
12	0.020	0.027	0.045
13	0.098	0.076	0.169
**rms**	**0.159**	**0.137**	**0.129**

**Table 9 molecules-28-00190-t009:** Structure Overlay calculations (Å) between the experimental {B_9_} fragment of YELXES and exemplar *klado*, *hypho* and *arachno* {B_9_} fragments.

Vertex	YELXES/*k*-{B_9_}	YELXES/*h*-{B_9_}	YELXES/*a*-{B_9_}
2	0.105	0.280	0.142
5	0.156	0.070	0.173
6	0.072	0.042	0.169
8	0.117	0.134	0.142
9	0.064	0.059	0.128
10	0.110	0.064	0.090
11	0.075	0.099	0.069
12	0.060	0.077	0.049
13	0.096	0.193	0.028
**rms**	**0.096**	**0.135**	**0.122**

**Table 10 molecules-28-00190-t010:** Structure Overlay calculation between the exemplar *hypho* {B_11_} fragment (left) and exemplar *klado* {B_11_} fragment (right).

	Atom	Misfit (Å)	
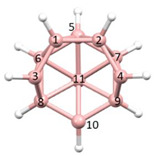	1	0.222	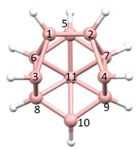
	2	0.222	
	3	0.355	
	4	0.355	
	5	0.066	
	6	0.099	
	7	0.098	
	8	0.055	
	9	0.055	
	10	0.251	
	11	0.025	
	**rms**	**0.201**	

**Table 11 molecules-28-00190-t011:** Structure Overlay calculations (Å) between the experimental {B_11_} fragment of BUPPEI (unprimed cage) and exemplar *hypho* {B_11_} fragment (left) and exemplar *klado* {B_11_} fragment (right).

	BUPPEI/*h*-{B_11_}			BUPPEI/*k*-{B_11_}		
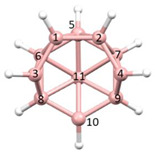	1	0.075	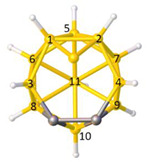	1	0.150	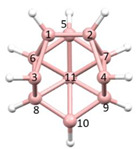
	2	0.062		2	0.165	
	3	0.094		3	0.267	
	4	0.103		4	0.258	
	5	0.034		5	0.095	
	6	0.059		6	0.047	
	7	0.060		7	0.046	
	8	0.043		8	0.023	
	9	0.031		9	0.037	
	10	0.052		10	0.222	
	11	0.012		11	0.015	
	**rms**	**0.062**		**rms**	**0.151**	

**Table 12 molecules-28-00190-t012:** Structure Overlay calculations (Å) between the experimental {B_11_} fragment of BUPPIM and exemplar *hypho* {B_11_} fragment (left) and exemplar *klado* {B_11_} fragment (right).

	BUPPIM/*h*-{B_11_}			BUPPIM/*k*-{B_11_}		
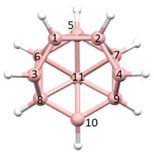	1	0.062	** 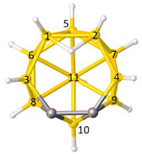 **	1	0.180	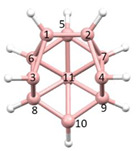
	2	0.064		2	0.175	
	3	0.062		3	0.317	
	4	0.054		4	0.324	
	5	0.021		5	0.086	
	6	0.050		6	0.050	
	7	0.051		7	0.052	
	8	0.028		8	0.042	
	9	0.026		9	0.043	
	10	0.050		10	0.273	
	11	0.016		11	0.023	
	**rms**	**0.047**		**rms**	**0.181**	

**Table 13 molecules-28-00190-t013:** Structure Overlay calculations (Å) between the {B_11_} fragment of DFT-optimized compound **1** (center) and exemplar *hypho* {B_11_} fragment (left) and exemplar *klado* {B_11_} fragment (right).

	1/*h*-{B_11_}			1/*k*-{B_11_}		
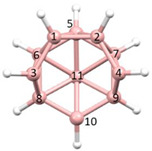	1	0.093	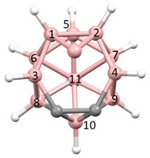	1	0.137	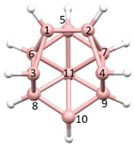
	2	0.090		2	0.138	
	3	0.169		3	0.188	
	4	0.168		4	0.189	
	5	0.017		5	0.059	
	6	0.055		6	0.051	
	7	0.056		7	0.051	
	8	0.025		8	0.031	
	9	0.026		9	0.031	
	10	0.098		10	0.163	
	11	0.024		11	0.005	
	**rms**	**0.091**		**rms**	**0.115**	

## Data Availability

The data presented in this study are available on request from the corresponding author.

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
