# Peer review of "Bridges and Vertices in Heteroboranes"

_molecules, 2022, doi:10.3390/molecules28010190_

Round 1

Reviewer 1 Report

This work presents an interesting approach for the classification of borane clusters containing non-boron atoms.  This reviewer believes that the strength of the authors' proposal would be increased if they provide commentary on the statistical significance (or estimates of reliability) for the rms differences that are computed: i.e. how different must the values be in order to determine that one cluster assignment is better than an alternative assignment.

One other comment is that the use of BP86/6-31G** for the computational aspects and generating the ideal models should probably be justified.  There are so many more modern methods and most approaches would typically employ dispersion corrections.

Overall, I think this is an interesting and valuable idea that merits publication.

Author Response

Response to Reviewer 1 Comments

Point 1: This work presents an interesting approach for the classification of borane clusters containing non-boron atoms.  This reviewer believes that the strength of the authors' proposal would be increased if they provide commentary on the statistical significance (or estimates of reliability) for the rms differences that are computed: i.e. how different must the values be in order to determine that one cluster assignment is better than an alternative assignment.

Response 1: A good point.  We have looked at a number of structure determinations of heteroboranes with crystallographically-independent molecules (i.e. Z’>1) and overlaid the {Bx} fragments of each molecule.  The rms misfit values are typically 0.02 Å or less (e.g. BUPPEI, {B11}, 0.015 Å; IPOBAS, {B9}, 0.020 Å; HIMMEX, {B10}, 0.011 Å.)  This also serves to confirm that for ZONCOU and FAFYAN, where the rms misfit is only 0.019 Å, the {B11} fragments are indeed isostructural.  In all the comparisons of computed {Bx} fragments of electronically-different fragments considered in the paper the rms misfit is 0.109 Å or greater.

Point 2: One other comment is that the use of BP86/6-31G** for the computational aspects and generating the ideal models should probably be justified.  There are so many more modern methods and most approaches would typically employ dispersion corrections.

Response 2: We have compared the structures of [B11H11]2— and [B12H12]2— computed with the BP86/6-31G** approach with the crystal structures of these species (UQOBAE and LOSDIG respectively in the CSD).  Rms misfit values based on the B atoms are 0.021 Å and 0.009 Å respectively, indicating good to very good agreement between experiment and computation.  Optimization with range of other functionals was tested but gave similar results (M06: 0.020 Å/0.008 Å; B97D: 0.023 Å/ 0.012 Å; TPSS: 0.017 Å/0.007 Å and wB97x-D: 0.016 Å/0.005 Å).  Recomputing the structure of [B12H12]2— at the BP86/6-311++G** basis set resulted in no appreciable change in the computed geometry (rms misfit between BP86/6-31g** and BP86/6-31G++g**= 0.0001 Å).

A comment to justify the choice of the BP86 functional has been added to the Computational Details section in the Supporting Materials.

Point 3: Overall, I think this is an interesting and valuable idea that merits publication.

Response 3: Many thanks.

Reviewer 2 Report

1) Authors should provide and rationalize the quntitative criteria for the RMS misfits to consider the fragments to be distinguishable. For example, in 3.1 section the RMS misfit value of 0.109 A was considered enough for distinguishing. But what if the RMS misfit value is 0.05 A? Or 0.02 A etc.?

2) Considering the equation (1) introducing the "verticity" parameter, it is not clear why this value (according to the authors statement) lies in the range of 0-100%. This statement should be proven, alternatively the proper interpretation of the "verticities" below 0 and above 100% should be provided. Back to the previous question, when the B/C RMS misfit approaching 0, the "verticity" will approach the infinity.

3) I am curiuos whether the attribution of the certain hetheroatom in the boron cluster to a bridge or vertecs has any correlation with the experimental properties of these clusters, or it is just the terminologic problem?

4) If the vertex vs bridge attribution can ba performed via an electron counting, at first glance the population analysis seems to be helpfull computational method. What are the advances of the "topological" criterion proposed by the authors over the "electronic" criterion derived from the population analysis?

5) The authors should provide an overview of the current approaches and criteria for distinguishing the vertex vs bridge in hetheroborane clusters. In the last paragraph (L419) authors mentioned some "conventional comparison" as an alternative of their criterion, although they provided no information about these "conventional comparison" approaches in the introduction or elsewhere.

Author Response

Response to Reviewer 2 Comments

Point 1: Authors should provide and rationalize the quantitative criteria for the RMS misfits to consider the fragments to be distinguishable.  For example, in 3.1 section the RMS misfit value of 0.109 A was considered enough for distinguishing.  But what if the RMS misfit value is 0.05 A?  Or 0.02 A etc.?

Response 1: Reviewer 1 made essentially the same point and our response is copied below:

A good point.  We have looked at a number of structure determinations of heteroboranes with crystallographically-independent molecules (i.e. Z’>1) and overlaid the {Bx} fragments of each molecule.  The rms misfit values are typically 0.02 Å or less (e.g. BUPPEI, {B11}, 0.015 Å; IPOBAS, {B9}, 0.020 Å; HIMMEX, {B10}, 0.011 Å.)  This also serves to confirm that for ZONCOU and FAFYAN, where the rms misfit is only 0.019 Å, the {B11} fragments are indeed isostructural.  In all the comparisons of computed {Bx} fragments of electronically-different fragments considered in the paper the rms misfit is 0.109 Å or greater.

Point 2: Considering the equation (1) introducing the "verticity" parameter, it is not clear why this value (according to the authors statement) lies in the range of 0-100%.  This statement should be proven, alternatively the proper interpretation of the "verticities" below 0 and above 100% should be provided.  Back to the previous question, when the B/C RMS misfit approaching 0, the "verticity" will approach the infinity.

Response 2: To illustrate consider PACBOR (although the same argument could be made for any of the species in the paper):

A is the experimental {B10} fragment, B is the computed arachno {B10} fragment and C is the computed hypho {B10} fragment.  Rms misfit B/C = 0.109 Å.

There are two extreme possibilities…

If A fits perfectly with the arachno fragment rms misfit A/B = 0 Å and rms misfit A/C = 0.109 Å.  Then verticity = {(0.109 + 0 – 0.109)/(2 × 0.109)} × 100% = 0%.

Alternatively, if A fits perfectly with the hypho fragment rms misfit A/B = 0.109 Å and rms misfit A/C = 0 Å.  Then verticity = {(0.109 + 0.109 – 0)/(2 × 0.109)} × 100% = 100%.

Point 3: I am curious whether the attribution of the certain heteroatom in the boron cluster to a bridge or vertex has any correlation with the experimental properties of these clusters, or it is just the terminologic problem?

Response 3: Our approach has simply been terminological – we are trying to find the most appropriate way to describe these species.

Point 4: If the vertex vs bridge attribution can be performed via an electron counting, at first glance the population analysis seems to be helpful computational method.  What are the advances of the "topological" criterion proposed by the authors over the "electronic" criterion derived from the population analysis?

Response 4: Electron counting cannot distinguish between a bridge or vertex, as there is no a priori means to assign a B to either role.  This is the advantage of our topological approach – it gives an unbiased assessment of bridge/vertex character that is based on the overall topology of the cluster.  In principle an analysis of the electronic structure may give insight into the extent of donation from a given B centre and hence imply a degree of verticity; however, given the highly delocalized nature of the electronic structure of these species and the subtle geometric trends they exhibit, we maintain that our topological approach is not only simpler to implement but also more robust in terms of the outcomes.

Point 5: The authors should provide an overview of the current approaches and criteria for distinguishing the vertex vs bridge in heteroborane clusters.  In the last paragraph (L419) authors mentioned some "conventional comparison" as an alternative of their criterion, although they provided no information about these "conventional comparison" approaches in the introduction or elsewhere.

Response 5: We note (lines 330-331) that to our knowledge there is no current method to distinguish between bridges and vertices.  In line 419 by ‘conventional’ we simply meant the parameters we listed previously (bond lengths, interbond angle, torsion angles etc).  In retrospect it would perhaps be clearer if we replaced “     ‘conventional’     “ with “     such conventional    “ and have done that in the revised manuscript.

Reviewer 3 Report

Jemmis' mno electron counting rule could be mentioned.

Please include BP86/631G** in Methods section.

Author Response

Response to Reviewer 3 Comments

Point 1: Jemmis' mno electron counting rule could be mentioned.

Response 1: Whilst the mno rule is useful for condensed polyhedral boranes it reduces to the Wade approach for single cluster species.  Since all the species we consider in the paper have only one polyhedron we see no real value in referencing the mno rule.

Point 2: Please include BP86/631G** in Methods section.

Response 2: Now included in revised manuscript.

Round 2

Reviewer 2 Report

All the points were adressed by the authors, now the manuscript can be accepted in present form.